METHODS

# *CLAW*: An automated Snakemake workflow for the assembly of chloroplast genomes from long-read data

**Aaron L. Phillips**[1☯]*, **Scott Ferguson**[2☯], **Rachel A. Burton**[1], **Nathan S. Watson-Haigh**[3,4,5]

**1** Department of Food Science, University of Adelaide, Adelaide, South Australia, Australia, **2** Research School of Biology, Australian National University, Canberra, Australian Capital Territory, Australia, **3** South Australian Genomics Centre (SAGC), SAHMRI, Adelaide, South Australia, Australia, **4** Australian Genome Research Facility, Victorian Comprehensive Cancer Centre, Melbourne, Victoria, Australia, **5** Alkahest Inc., San Carlos, California, United States of America

☯ These authors contributed equally to this work.
* aaron.phillips@adelaide.edu.au

**Data Availability Statement:** All data used in this study were publicly available from NCBI. The ONT long read accession numbers accessed for this study via NCBI were: ERR3237140, DRR149372, ERR5421724, DRR252953, SRR9643837,

## Abstract

Chloroplasts are photosynthetic organelles in algal and plant cells that contain their own genome. Chloroplast genomes are commonly used in evolutionary studies and taxonomic identification and are increasingly becoming a target for crop improvement studies. As DNA sequencing becomes more affordable, researchers are collecting vast swathes of high-quality whole-genome sequence data from laboratory and field settings alike. Whole tissue read libraries sequenced with the primary goal of understanding the nuclear genome will inadvertently contain many reads derived from the chloroplast genome. These whole-genome, whole-tissue read libraries can additionally be used to assemble chloroplast genomes with little to no extra cost. While several tools exist that make use of short-read second generation and third-generation long-read sequencing data for chloroplast genome assembly, these tools may have complex installation steps, inadequate error reporting, poor expandability, and/or lack scalability. Here, we present *CLAW* (Chloroplast Long-read Assembly Workflow), an easy to install, customise, and use Snakemake tool to assemble chloroplast genomes from chloroplast long-reads found in whole-genome read libraries (https://github.com/aaronphillips7493/CLAW). Using 19 publicly available reference chloroplast genome assemblies and long-read libraries from algal, monocot and eudicot species, we show that *CLAW* can rapidly produce chloroplast genome assemblies with high similarity to the reference assemblies. *CLAW* was designed such that users have complete control over parameterisation, allowing individuals to optimise *CLAW* to their specific use cases. We expect that *CLAW* will provide researchers (with varying levels of bioinformatics expertise) with an additional resource useful for contributing to the growing number of publicly available chloroplast genome assemblies.

SRR13908657, DRR196880, SRR11472010, SRR9858982, ERR3850904, ERR4852503, SRR8692273, SRR10377593, ERR3374012, SRR10194526, SRR13070229, ERR3430399, SRR12407219, SRR12549534. The PacBio long read accession numbers accessed for this study via NCBI were: SRR21973883, DRR316159, ERR8705848, DRR075367, ERR11472546, SRR8517588, SRR8892931, SRR10189116, SRR6335233, SRR19732304, SRR6656266, SRR16267434, SRR9994113. The complete chloroplast reference genome assembly accession numbers used for this study via NCBI were: NC_005353, NC_015359, NC_008289, NC_012097, NC_034777, NC_023533, NC_008155, NC_015891, NC_029243, NC_027223, NC_031855, NC_022393, NC_023216, NC_014063, NC_003119, NC_006290, NC_034696, NC_028069, NC_013843. The code for CLAW and a test dataset from NCBI can be found at https://github.com/aaronphillips7493/CLAW.

**Funding:** The author(s) received no specific funding for this work.

**Competing interests:** The authors have declared that no competing interests exist.

## Author summary

Chloroplast genomes are important resources as they can be used to help resolve phylogenies and aid in species identification. The importance of chloroplasts and their genes in algal and plant stress responses is a field of research in its infancy that stands to benefit greatly from increased publicly available chloroplast sequence data. As long-read sequencing technology becomes more accessible, researchers can generate, and access troves of data contained in long-read libraries. Often embedded in these libraries are chloroplast reads. With the right tools, these reads can be extracted and used for chloroplast genome assembly. For novice users, existing tools can be hard to install, requiring multiple manual steps, have poor reporting of errors when they occur, have poor expandability, and/or lack scalability. Together, these features can reduce accessibility to non-expert users. Here, we present *CLAW* (Chloroplast Long-read Assembly Workflow)–an easy to install, easy to use workflow for the assembly of chloroplast genomes from long-read data. We anticipate that this new tool will lower barriers to entry that might dissuade novice users from participating in the field of bioinformatics and encourage the *de novo* assembly of chloroplast genomes from diverse algal, plant, and other photosynthetic species.

This is a *PLOS Computational Biology* Methods paper.

## Introduction

Chloroplasts are organelles that perform photosynthesis in photosynthetic cells. They convert light energy into a stable form of chemical energy, a process essential to life on earth. In addition to their importance as the organelle of photosynthesis, chloroplasts have been extensively used throughout biological research in part because they contain their own DNA genome.

The typical chloroplast genome is circular with a quadripartite structure comprised of a large single copy (LSC) region, a small single copy (SSC) region, and two inverted repeats (IR) [1]. Chloroplast genomes have low mutation rates and are highly conserved, usually maternally inherited, do not undergo recombination, and are typically 120–160 kbp in length [2]. Each photosynthetic cell contains many chloroplast organelles, and each chloroplast can contain 10,000 or more copies of its genome [3–6]. All these features make chloroplast genomes well-suited to studies in phylogenetics [7,8] and species identification [9,10]. There is also growing interest in the role(s) that chloroplasts play in responses to stress and thus how they can contribute to, for example, enhancing crop performance in a changing world [11,12].

Most chloroplast genome assembly efforts to date have relied on short-read (100–150 bp) sequencing technologies, such as Illumina. As such, several short-read specific chloroplast assembly tools have been developed [13]. While short reads are highly accurate, they often fail to assemble repetitive genomic regions [14] and can fail to detect the structural variations that are now known to be pervasive in genomic sequences (e.g., [15]). Assembly of chloroplast genomes, despite their small size, also suffers from this phenomenon because of the IR regions [16,17]. Additionally, the LSC and SSC regions of the chloroplast can be challenging for short-read assemblers to resolve because these regions can exist in a 'flip-flop' state–a term used to describe the tendency of different regions (usually the SSC or IRs) of the genome to invert, leading to the possibility of multiple chloroplast genome assemblies from one cell [17–19].

With the advent of third generation long-read sequencing (e.g., Oxford Nanopore Technology (ONT), Pacific Biosciences), we can now affordably generate reads that span or can assemble across large repeats. Long-read sequencing allows for computationally easier and more structurally accurate assembly of genomes [20]. It is becoming increasingly feasible for research groups of any size to collect large volumes of whole-genome sequencing data for their species of interest. Although whole-genome sequencing projects target the nuclear genome, many chloroplast sequencing reads will typically be generated as a side effect since the chloroplast genome exists at high copy numbers within tissues [5]. Thus, if photosynthetic tissue is used to generate reads, chloroplast genomes can be assembled without any additional sequencing.

Different chloroplast genome assembly methods have been developed [13]. All methods share two key steps: 1) identification and extraction of chloroplast reads; and 2) assembly of the chloroplast genome. The identification of chloroplast reads from whole-genome read libraries can be achieved by k-mer analysis, where reads containing k-mers within a specific frequency range are assumed to be of plastid origin [21]. Alternatively, reads can be extracted from read libraries by similarity to a reference sequence–only those reads that aligned to a reference sequence are used for genome assembly; this process is called 'read baiting'. Here, we present *CLAW*, which uses the read baiting method to assemble chloroplast genomes with circular-sequence-aware assemblers *Flye* [22] and *Unicycler* [23].

Despite the increasing use of long-read sequencing technologies, an easy to use, automated, and reliable method to assemble long-reads into chloroplast genomes remains unavailable. *CLAW* is an easy to install and easy to use tool for the reference-guided long-read assembly of chloroplast genomes that was designed using best-practice principles [24]. This tool provides users having little bioinformatics experience a fast, easy, and reproducible way of assembling chloroplast genomes from long-reads. Our method is automated, requiring only minimal user input and makes use of freely available and/or published tools.

## Methods

### Overview of *CLAW*

*CLAW* begins with a long-read whole genome sequencing library, stored in either the FASTA or FASTQ format and optionally *gzip* compressed (Fig 1). If using Oxford Nanopore Technologies (ONT) technology, the raw sequencer output (fast5, or the newly developed POD5 format) must be base-called by base-calling software (e.g., *Guppy*, *Dorado*, or another method). As *CLAW* makes use of a reference chloroplast genome (RCG) to bait reads for assembly, it is a requirement to provide a RCG of the focal species or a closely related species. *CLAW* will download this reference—the user must only provide a National Center for Biotechnology Information (NCBI) reference sequence identifier (e.g., NC_031333.1). Additional parameters (See 'User Specifications' below) can be set to specify whether the user is working with FASTA or FASTQ file(s), the kind of long-reads the user is working with (e.g., ONT vs PacBio), the number of aligned reads to subsample for assembly, the expected chloroplast genome size, and how many CPUs *CLAW* should use.

After downloading the RCG, *CLAW* circularises (i.e., duplicates the reference sequence and then joins the 3' end of one to the 5' end of the other; Fig 2) the RCG to facilitate read mapping with *minimap2* (using the command: minimap2 -ax {config[minimap2_parameter]} -t {threads} {input.reference} {input.fastFile}, where 'a' generates CIGAR in the SAM format, 'x' is used to specify the read format; *CLAW* queries config.yaml for user-specified read format and thread value). This step is required as chloroplast genomes, which are circular in nature, are stored as linear sequences. Attempting to map reads across the break point may result in

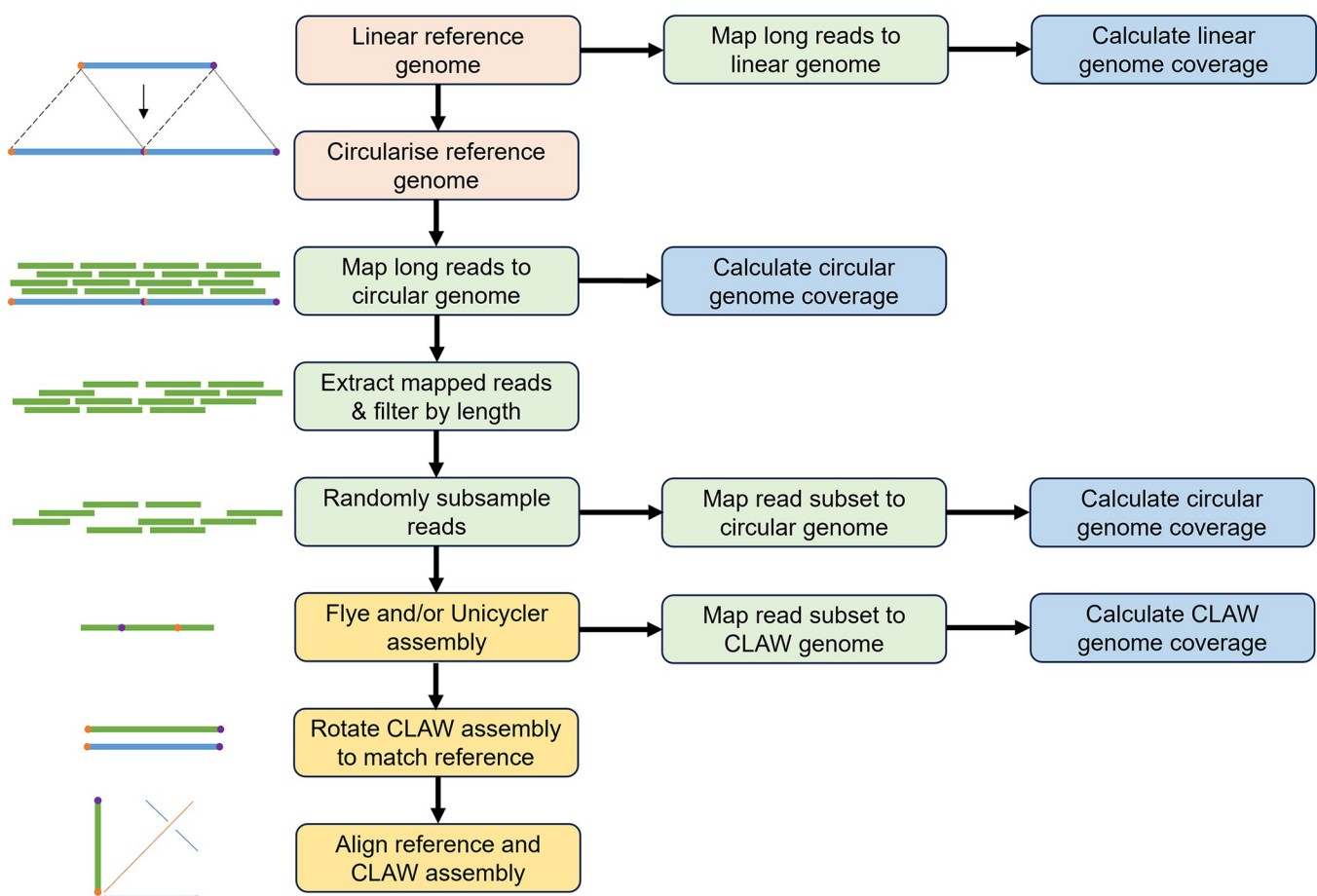

**Fig 1. Graphical representation of the *CLAW* workflow.** A linear reference genome is circularised (see Fig 2). Long reads (ONT or PacBio) are mapped to the circularised reference genome. Mapping reads are filtered for length and quality, then a random subsample of these reads are used for genome assembly via *Flye* and/or *Unicycler*.

poor read alignment (Fig 2). The read library is mapped to the RCG, and all chloroplast mapping reads (CMRs) are extracted. The extracted CMRs are filtered to remove reads shorter than a user-defined threshold (default: 5 kbp) in length and reads that are larger than the expected chloroplast genome size. Extremely high coverage can confound assemblers and worsen results, as such a reduced subset of reads is randomly selected for assembly. For repeatability, a seed for read sub-setting can be specified or, alternatively, a random seed will be generated by *CLAW*. The number of randomly selected assembly reads can be user-specified, enabling easy coverage adjustment for tuning of assembly time and success (e.g., more reads will give higher genome coverage but will increase the time required for assembly) and users can define a read mapping quality if required.

Assembly is performed using *Flye* and/or *Unicycler*, resulting in the generation of a FASTA file (containing the assembly) and a genome graph (Graphical Fragment Assembly (GFA) file, showing paths through the genome). For assessment of chloroplast assembly, potential chloroplast sequences will be rotated to match the breakpoint of the reference chloroplast genome and a dot plot produced using the MUMmer3 suite [25]. Additionally, *CLAW* will also produce BAM and bigwig files of read alignments to RCG and the *CLAW*-generated chloroplast genome assembly such that read depth can be investigated. If a complete chloroplast genome is not produced, *CLAW* can be rerun with a different random seed, and different number of

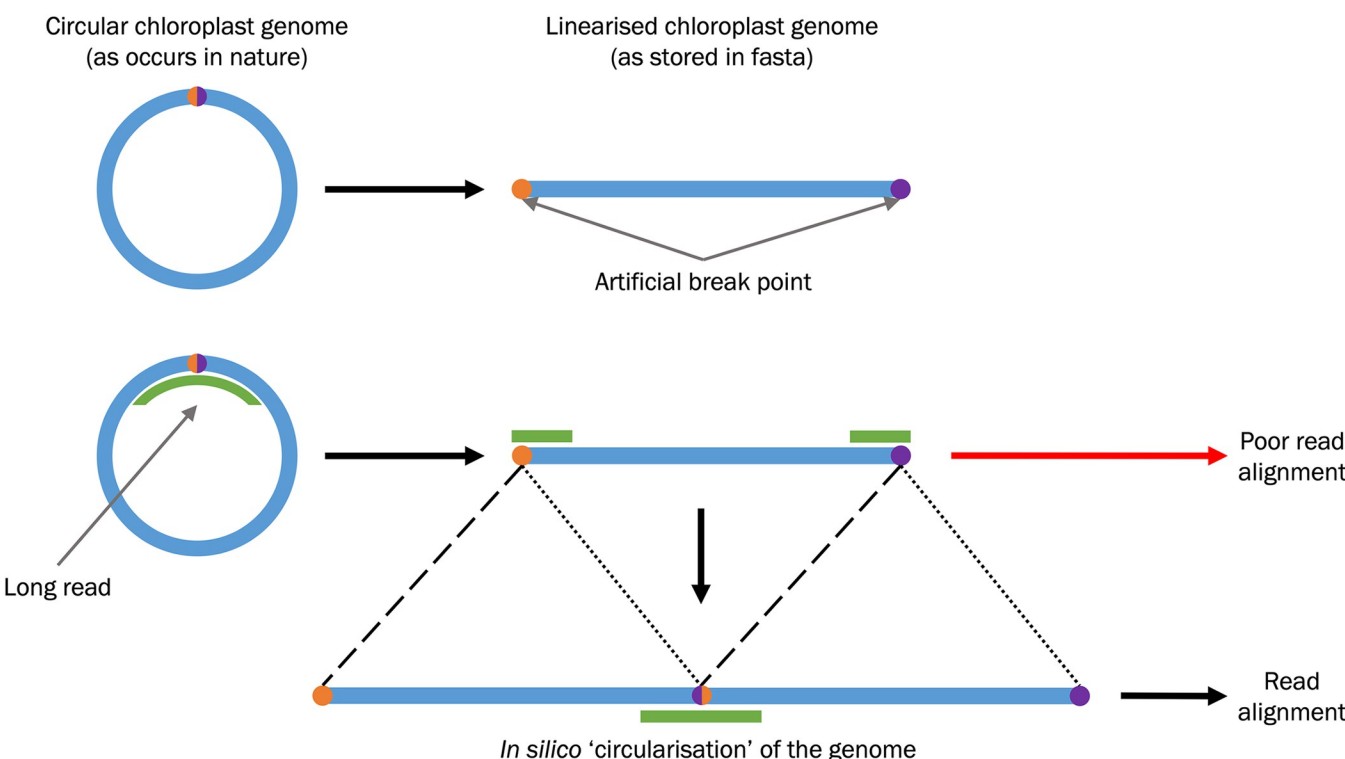

**Fig 2. Demonstration of 'circularising',** *in silico*, **linear reference chloroplast genome sequences after download from online databases.** Linearised chloroplast genome sequences introduce artificial breakpoints in the sequence (orange and purple circles). Artificial breaks may lead to poor long read (green lines) alignment, which may affect chloroplast read enrichment/baiting. *In silico* re-circularisation of the linear sequence may allow long reads to map across the artificial break points.

assembly reads (higher or lower coverage). The user is also encouraged to investigate potential sources of error via the log files generated as part of the workflow.

## User specifications

For *CLAW* to operate successfully, the user must edit the "config.yml" file located in the same directory as the Snakefile. Within "config.yml" the **mandatory** user-definable fields are ncbi_reference_accession, my_email, fast_file, flye_parameter, minimap2_parameter, and chloroplast_size.

There are also some editable parameters that users need not change. While these parameters need not be edited, the assembly process can be optimised on a per sample basis by fine tuning them: rand_seed, number_reads, read_min_length, read_quality, and cpus. If *CLAW* fails to produce a quality chloroplast genome assembly, users are advised to adjust either "rand_seed", "number_reads" and/or "read_quality" and re-run the workflow.

## Testing *CLAW*

We used publicly available ONT long-read libraries generated from 19 species (4 algal, 4 monocot, and 11 eudicots) with previously assembled chloroplast genome sequences to test *CLAW* (Table 1) and PacBio long-read libraries generated from 13 of the same 19 species that had publicly available data (S2 Table). Long-read libraries were downloaded from NCBI using *fasterq-dump* or from ENA using *axel* and saved into a directory called "chloro_assembly/reads" within the cloned git repository. Reference genomes were downloaded from NCBI

**Table 1. Information on ONT longreads used as input for *CLAW* and the *Flye*-generated chloroplast genome assembly statistics.**

| Genome graph in Fig 3 | Taxonomic group | Species | Long read accession no. | Long reads used as input (Mbp) | Reference chloroplast genome accession no. | Reference chloroplast size (kbp) | Assembly size (kbp) | | No. contigs | | Mean coverage (x) | | Similarity (%) | Time to completion (min) | RAM used (Gb) |
|---|---|---|---|---|---|---|---|---|---|---|---|---|---|---|---|
| | | | | | | | Chl | Mit | Chl | Mit | Chl | Mit | | | |
| A | Algae | *Chlamydomonas reinhardtii* | ERR3237140 | 60 | NC_005353 | 204 | 204 | - | 2 | - | 240 | - | 98.2 | 15.2 | 15.9 |
| B | Algae | *Chlorella variabilis* | DRR149372 | 21 | NC_015359 | 124 | 120 | - | 1 | - | 161 | - | 96.2 | 5.2 | 10.9 |
| C | Algae | *Ostreococcus tauri* | ERR5421724 | 8.5 | NC_008289 | 72 | 75 | 62 | 1 | 1 | 92 | 35 | 99.5 | 3.3 | 9.8 |
| D | Algae | *Pycnococcus provasolii* | DRR252953 | 12 | NC_012097 | 80 | 80 | - | 1 | - | 136 | - | 99.5 | 4.9 | 9.4 |
| E | Monocot | *Asparagus officinalis* | SRR9643837 | 22 | NC_034777 | 157 | 136 | - | 2 | - | 107 | - | 99.1 | 8.1 | 12.0 |
| F | Monocot | *Deschampsia antarctica* | SRR13908657 | 33 | NC_023533 | 135 | 139 | - | 1 | - | 133 | - | 99.1 | 5.0 | 10.8 |
| G | Monocot | *Oryza sativa* | DRR196880 | 42 | NC_008155 | 135 | 163 | - | 1 | - | 230 | - | 99.4 | 12.9 | 15.8 |
| H | Monocot | *Spirodela polyrhiza* | SRR11472010 | 67 | NC_015891 | 169 | 168 | - | 1 | - | 249 | - | 99.2 | 19.8 | 17.5 |
| I | Dicot | *Aquilaria sinensis* | SRR9858982 | 54 | NC_029243 | 160 | 180 | - | 1 | - | 233 | - | 98.4 | 9.5 | 15.7 |
| J | Dicot | *Cannabis sativa* | ERR3850904 | 42 | NC_027223 | 154 | 189 | - | 1 | - | 199 | - | 99 | 8.9 | 14.7 |
| K | Dicot | *Corylus avellana* | ERR4852503 | 26 | NC_031855 | 160 | 189 | - | 3 | - | 109 | - | 99.6 | 5.9 | 12.4 |
| L | Dicot | *Eucalyptus polybractea* | SRR8692273 | 36 | NC_022393 | 160 | 158 | - | 2 | - | 176 | - | 98.8 | 7.0 | 13.8 |
| M | Dicot | *Gossypium longicalyx* | SRR10377593 | 41 | NC_023216 | 160 | 175 | - | 2 | - | 171 | - | 98 | 6.3 | 13.5 |
| N | Dicot | *Lathyrus sativus* | ERR3374012 | 67 | NC_014063 | 121 | 120 | - | 1 | - | 209 | - | 99.2 | 7.5 | 13.7 |
| O | Dicot | *Medicago truncatula* | SRR10194526 | 45 | NC_003119 | 124 | 123 | - | 1 | - | 287 | - | 99 | 15.4 | 15.9 |
| P | Dicot | *Panax ginseng* | SRR13070229 | 29 | NC_006290 | 156 | 201 | - | 3 | - | 118 | - | 99.6 | 6.3 | 11.9 |
| Q | Dicot | *Prunus dulcis* | ERR3430399 | 31 | NC_034696 | 158 | 162 | - | 1 | - | 142 | - | 99.1 | 6.7 | 14.5 |
| R | Dicot | *Solanum commersonii* | SRR12407219 | 46 | NC_028069 | 156 | 175 | - | 2 | - | 221 | - | 99.4 | 9.3 | 11.3 |
| S | Dicot | *Vigna radiata* | SRR12549534 | 76 | NC_013843 | 151 | 150 | - | 1 | - | 316 | - | 98.8 | 11.3 | 16.9 |

using the *entrez-direct* 'esearch' function and saved into a directory called "chloro_assembly/reference" within the cloned git repository. Resources for each job in *CLAW* (e.g., time and memory) were defined in "cluster-configs/default.yaml" and tracked using Snakemake's internal "benchmark" feature. *CLAW* was executed on a Slurm High Performance Computer (HPC) by running the following command for each sample: "snakemake—profile profiles/slurm–use-conda chloro_assembly/{sample}~{assembler}_chloroplast.fasta". Where {sample} is the name of the read file deposited in "chloro_assembly/reads" and {assembler} can be "flye" or "unicycler". Each time *CLAW* was run, the reference genome for each respective read file was specified in the "config.yaml" file. If the user has multiple sample files from the same genera, or a closely related taxonomic group, in the "chloro_assembly/reads" directory, *CLAW* will attempt to assemble a genome for each using the same reference genome for read baiting. *DNAdiff* [25] was used to calculate percent identity between the assembled chloroplast genomes and their corresponding reference genome assemblies. While the analyses presented here were performed on a HPC with a Slurm workload management system in place, users should be aware that *CLAW* is perfectly capable of running on a local device or with other workload management systems (please see README for details).

We have supplied a subset (~39.8 Mbp) of an ONT read library generated from domestic rice tissue (*Oryza sativa cv*. IR64 –an *indica* rice; DRR196880) as well as a reference *O. sativa*

chloroplast genome (NC_008155.1) for users to test their installation of *CLAW*. This should allow users to assemble a rice chloroplast genome of ~136 kbp in length with ~290x coverage.

## Exploring the assemblies

GFA files generated by *Flye* were used as input for *Bandage* [26] to visualise genome structure and confirm the assemblies as chloroplast sequence as follows. The coding sequences of all chloroplast and mitochondrial-encoded genes for the test samples (Table 1) were downloaded from NCBI as FASTA files. After building a BLAST database for each chloroplast assembly in *Bandage*, the chloroplast and mitochondrial coding sequences were used to annotate each assembly using the BLAST annotation feature within *Bandage*. Mitochondrial coding sequences were used to test whether the assembled contigs were of mitochondrial origin.

The Rubisco large subunit (RbcL) genomic sequences were identified in each of *CLAW*'s assemblies by using the *BLAST+ suite* [27] and publicly available RbcL sequences from each of the reference chloroplast genomes as the query. The extracted RbcL genes were aligned to their respective reference RbcL genes in *UGENE* to assess sequence differences (manual inspection). RbcL genes extracted from the reference and *CLAW*-assembled genomes were used as input for a Neighbour-Joining global alignment tree with free end gaps using the Tamura-Nei genetic distance model with no outgroup and a 5.0/4.0 cost matrix in Geneious Prime.

## Results

Using 19 publicly available ONT long read libraries, 13 publicly available PacBio long read libraries, and their corresponding reference chloroplast genome assemblies, we show that *CLAW* can be used to assemble chloroplast genomes from long reads of chloroplast origin contained within whole genome shotgun long read libraries (Fig 3 and Tables 1, S1, and S3). For both long read technologies, *Unicycler* workflows completed faster than *Flye* workflows and used less RAM than *Flye* workflows (Tables 1, 2, S1, and S3). *Flye*-assembled genomes were more similar to their reference genomes than *Unicycler*-assembled genomes. For the ONT data, *Flye* assemblies were more contiguous than those generated by *Unicycler*, though the opposite was true for the PacBio data. Given the similarities in the assemblies generated using ONT and PacBio data, for simplicities sake, we focus on the results from the ONT *Flye*-generated assemblies here. The mean finish time of all 19 ONT *Flye* tests of *CLAW* was 8.9 mins, with a range of 3.3–19.8 mins (Table 2). The read alignment steps using *minimap2* and the chloroplast genome assembly by *Flye* consumed the most resources out of all jobs submitted as part of *CLAW*, with assembly requiring an average time of 5.3 mins and 8.3 Gb (RAM) to complete, and initial read alignment an average of 1.9 mins and 2.6 Gb.

When using ONT data and *Flye*, *CLAW* was able to assemble 11 of the 19 (~58%) chloroplast genomes into a single contig (Table 1). Alignment of the assembled chloroplast genomes to their corresponding reference genomes produced the canonical chloroplast-chloroplast genome alignment pattern (Fig 3). Table 1 shows high sequence similarity (mean = 98.9%) between the genomes assembled by *CLAW* and the reference chloroplast genomes. Assembled genome size was on average ±15.5 kbp different from the reference chloroplast genome size, with a range of 0–65 kbp (Table 1). Ten of the assembled genomes were, on average, 20.2 kbp larger than the expected reference genome size, and seven were, on average, 4.4 kbp smaller than the expected reference genome size, while two assemblies were the expected size. *Bandage* annotation using publicly available coding-gene information for each of the reference genomes shows that all of the assemblies are of chloroplast genomes (except for two additional sequences; see below). The Bandage plots also show a mixture of genome structure

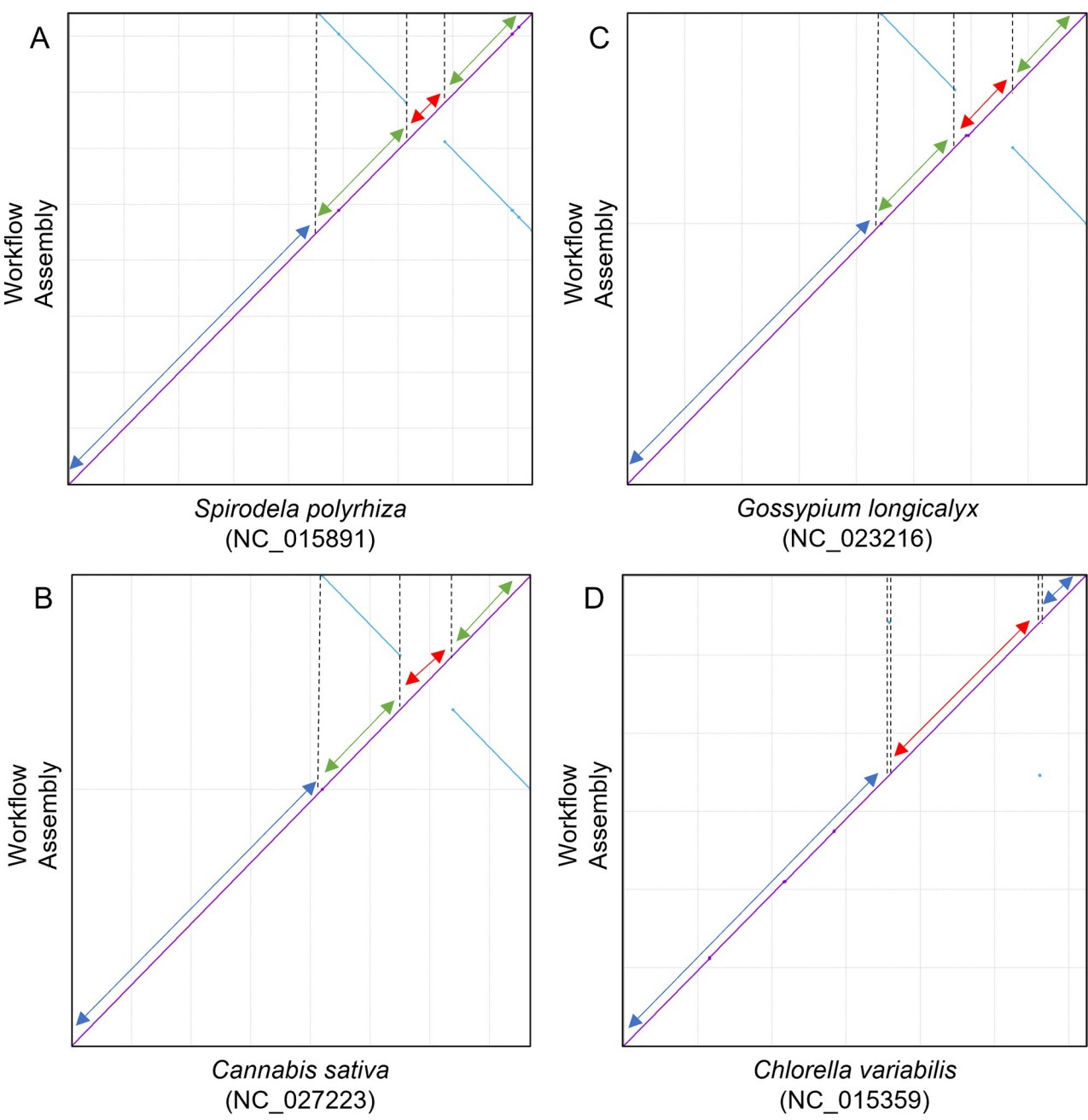

**Fig 3.** Representative reference-sample chloroplast genome alignments for (A) a monocot species, (B & C) two dicot species, and (D) an algal species. The reference genome is represented on the X-axis, and the genome assembled by *CLAW* is represented on the Y-axis. Species names and reference genome NCBI accession numbers appear on the X-axis. Please refer to Table 1 for the ONT long read accession numbers used by *CLAW* for genome assembly using *Flye*. Each alignment presented here follows the canonical chloroplast genome-genome alignment patterns and the LSC (dark blue line in A), SSC (red line in A), and the two IR (green lines in A) regions are clearly identifiable (broken black line in A indicates the boundaries of each region).

representations (Fig 4). Due to the sequence similarity of the IR sequences, resulting assembly graphs may be represented as lassos (IRs sequences are more similar) or as circles (IR sequences are less similar). For the algal samples, 50% (2/4) of the assemblies are represented as closed circles, and the remaining 50% are represented as a lasso-like structure (Fig 4). As expected, *CLAW* was also able to assemble part of a mitochondrial genome for some of the

**Table 2. Mean (± SE) time to completion, RAM used, percent identity to the reference genomes, and number of contigs generated by *CLAW* following the *Flye* and *Unicycler* workflows with ONT or PacBio data as input.** All jobs were run on Intel X86-64 Haswell and Skylake CPUs.

| Long reads | Assembler | Time (min) | RAM (Gb) | Percent identity to reference (%) | No. contigs |
|---|---|---|---|---|---|
| **ONT** | *Flye* | 8.87 ± 0.99 | 13.5 ± 0.56 | 98.9 ± 0.18 | 1.5 ± 0.16 |
| | *Unicycler* | 6.19 ± 0.73 | 6.1 ± 0.62 | 98.6 ± 0.18 | 3 ± 0.44 |
| **PacBio** | *Flye* | 10.6 ± 1.21 | 7 ± 0.77 | 98.9 ± 0.77 | 3 ± 1.25 |
| | *Unicycler* | 8.8 ± 1.23 | 6.2 ± 0.66 | 98.3 ± 0.75 | 1.4 ± 0.15 |

tested samples (in this case only the *Flye* assembly of *Ostreococcus taurii*; Fig 4C). Users will need to be cognisant that this could occur for their samples too and may inflate the total assembly size. For example, the *O. tauri* assembly consisted of two contigs with a total size of 137 kbp. However, the expected chloroplast genome size was 72 kbp (Table 1). Identification and removal of the mitochondrial contig(s) left one contig totalling 75 kbp in length. For the monocot samples, all four of the assemblies are represented by the "lasso" structure. For the eudicot samples, ~45% (5/11) of the assemblies are represented as closed circles, and the remaining ~55% are represented as "lasso's". For *Corylus avellana*, a second sequence that represents the IR for this species is present in the bandage plot (Fig 4K).

We identified and extracted the plastid sequence of the RbcL genes from the 19 ONT *Flye* assemblies. On average, these genes were 99.3% identical to their corresponding reference RbcL gene sequences, with a range of 98–100% identity (see S1 File for example alignments between reference and *CLAW*-generated RbcL sequences; S1 Fig). In total, there were 106 indels and 16 substitutions (122 total variations) between assembled RbcL genes and their reference sequences. Excluding the 106 indels, 17 of the 19 assembled RbcL genes had 100% sequence identity to their corresponding reference sequences. The RbcL genes from the remaining 2 assemblies (*O. taurii*, and *Chlorella variabilis*) had ~99% sequence identity to their respective reference sequences due to the presence of the 16 substitutions.

## Discussion

Here, we present *CLAW*—an easy to install, easy to customise, highly scalable, and easy to use tool for assembling chloroplast genomes from long reads identified and extracted from Whole-Genome Sequencing (WGS) data sets. We tested *CLAW* using both ONT and PacBio datasets. Pre-existing chloroplast genome assembly tools largely make use of short read data [13]. However, Zhou et al. [28] introduce ptGAUL as a plastid genome assembly tool that makes use of long reads. While ptGAUL and *CLAW* share the common objective of chloroplast genome assembly from long reads their implementation methodologies markedly differ. ptGAUL operates as a single, large shell script and lacks robust error handling and recovery capabilities. In contrast, *CLAW* leverages a Snakemake workflow, a modular and flexible framework that enhances reproducibility. The utilisation of Snakemake in *CLAW* empowers scalability, facilitating efficient processing of large datasets, a feature not inherently supported by ptGAUL's single-script design. Therefore, the modular and adaptable nature of *CLAW*'s Snakemake workflow distinguishes it as a more versatile, user-friendly, and scalable tool compared to ptGAUL. Furthermore, Jin et al. [29] report on the possibility of using long read data to assemble chloroplast genomes with their bespoke tool, *GetOrganelle*. However, at the time of writing, chloroplast genome assembly using long read data is not implemented with the *GetOrganelle* toolkit. We show that *CLAW* can be used to assemble high quality chloroplast genomes from chloroplast reads from within whole genome sequencing data. *CLAW* can be used to glean additional value from any WGS project targeting photosynthetic tissues/cells (e.g., leaves, or algal cell suspensions).

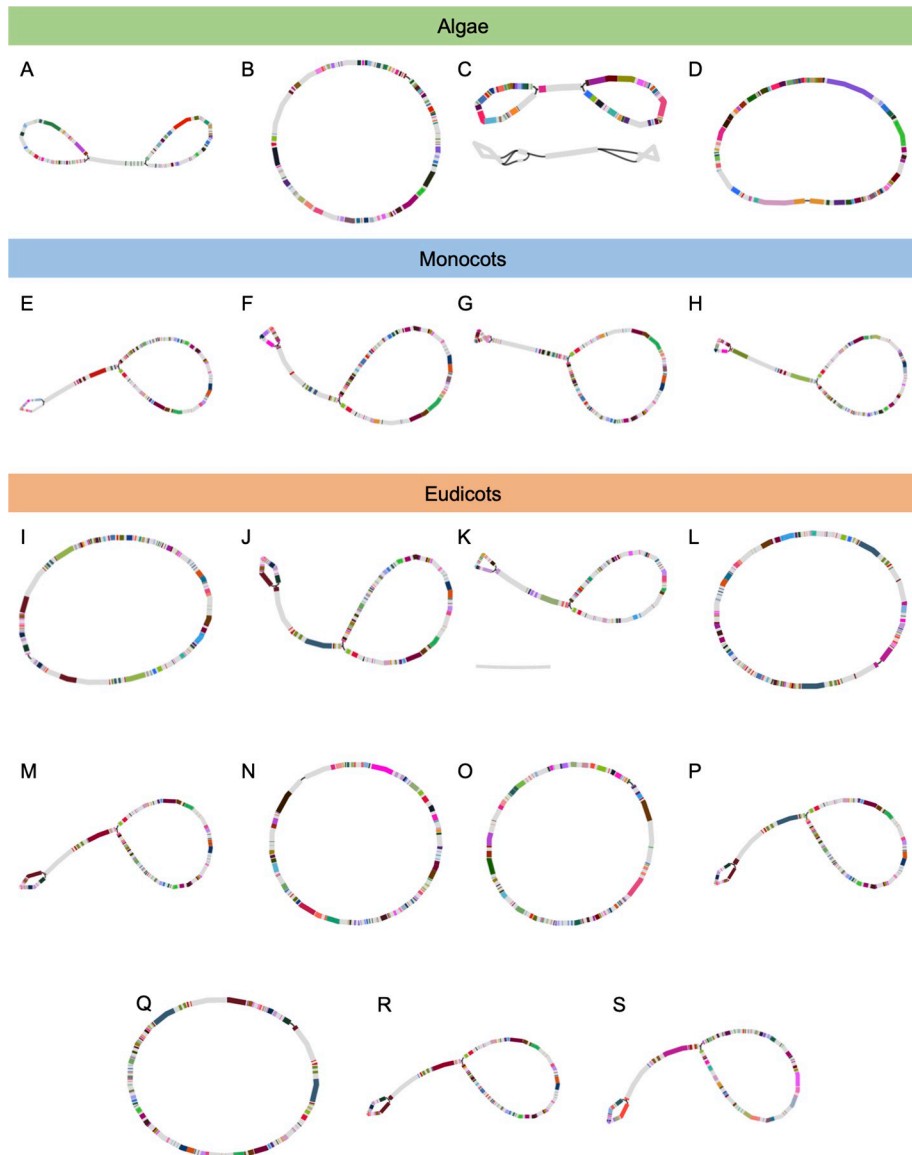

**Fig 4. Repeat graphs for the 19 chloroplast genomes (4 algal species, 4 monocot species, and 11 eudicot species) assembled by *CLAW* using ONT long reads as input for *Flye*.** The 'lasso' style genome plots represent assemblies in which the IR regions (the flat lines connecting two circular pieces) are perfectly palindromic in the assembly, while the circular style genome plots represent assemblies in which the IR regions are not identical. The order of species in this plot follows the order of species in Table 1. The colouring of segments of each genome represents genome annotations assigned using the BLAST-based genome annotation feature of *Bandage*. Genomes were annotated using publicly available coding region annotations from each of the reference chloroplast genomes. (C) and (K) have additional contigs that could not be annotated using chloroplast coding regions as they are mitochondrial genome fragments.

A recent review of chloroplast genome assembly tools making use of short read data showed a large range of completion times for genome assembly (~3 to ~835 mins; 13). Further, the peak memory required for the analysed tools/parameters was ~2.6–25.2 Gb. Jin et al. [29] compared *GetOrganelle* to another plastid assembly pipeline, *NOVOPlasty*, and found that run-times varied substantially (~4 to ~1,024 mins), with a peak memory consumption of ~2–4 Gb. For the ONT samples, it took *CLAW* an average of ~9 mins and ~14 Gb to assemble each of the 19 chloroplast genomes. Thus, the pipeline reported here is time efficient, though more

memory intensive than some assembly tools making use of short read data. Similar results were obtained for the PacBio samples. Given that *CLAW* makes use of long-read data, this is to be expected. However, we acknowledge that comparison of computation times across different hardware presents some challenges.

*CLAW* was able to assemble chloroplast genomes for all 19 test ONT datasets and the 13 PacBio test datasets (*Flye* average identity = 98.9%). The accuracy of the assemblies relative to the reference genomes was not as high as the accuracy obtained from short read assemblers (which achieved an average identity of 99.9%, 29). This is to be expected for ONT long reads, as short reads typically have higher per base accuracy than ONT long reads (cv. 0.09–0.47% error rate in Illumina short reads vs. ~2% error rate for ONT long reads depending on the flow cell and base caller used; [30–33]). However, high coverage, such as those achieved by *CLAW* for the assembled genomes (mean coverage = 183x), can reduce the impact of random (though not systematic) ONT errors on the assembly. Assemblies made using PacBio and HiFi reads are likely to have fewer residual assembly errors due to a more uniformly distributed error profile or higher base-level accuracy than ONT [34,35]. However, we found that our PacBio *Flye* assemblies were just as accurate to the reference chloroplast genomes as the ONT *Flye* assemblies. It is important to note that the reference genomes and the read libraries used to test *CLAW* came from different individuals within the same species. Thus, users might expect to see deviation from the reference genome in the *CLAW*-assembled genomes. Users may consider the use of short read technology, if available, or nanopore reads with enough sequencing depth, to correct errors in their assemblies [36].

While we report high accuracy of the assembled chloroplast genomes, *CLAW* also produced incomplete, fragmented assemblies for some of the mitochondrial genomes (Fig 4 and Tables 1, S1, S2 and S3). The in-built BLAST-based annotation feature of *Bandage* confirmed that the assembled chloroplast genomes were indeed chloroplast genomes and that the other sequences sometimes assembled were of mitochondrial origin. *CLAW* probably attempted to assemble mitochondrial sequences because some mitochondrial reads were likely included in the assembly read pool due to high sequence similarity with the reference chloroplast genomes. For example, Zhang et al. [37] report high (up to 72%) sequence similarity between plastid and mitochondrial sequences. Further, maize (*Zea mays*) mitochondrial and chloroplast genomes contain a 12kbp stretch of sequence that is identical [38]. Sequence similarity between mitochondrial and chloroplastic genomes comes about in part due to ancient chloroplast-to-mitochondria gene transfer events [39]. Thus, the read-enrichment strategy employed by *CLAW* is capable of baiting mitochondrial reads as well as chloroplastic reads. Future work should aim to either reduce or increase the representation of mitochondrial reads in the baited read set to avoid aberrant mitochondrial genome assemblies, or to make it possible to assemble more complete mitochondrial genome sequences, respectively. Reducing the representation of mitochondrial reads in the baited read library may be possible by implementing a filtering rule that uses sequence similarity with publicly available mitochondrial genome sequences. A k-mer approach could also be used to delineate chloroplastic and mitochondrial reads in the future.

We were able to identify and isolate RbcL genes from all of the chloroplast genomes assembled using *CLAW*. Our PacBio *Flye* assemblies contained RbcL genes with and average identity of 99.9% to the reference sequences (S1 File and S1 Fig). The RbcL genes derived from our ONT *Flye* assemblies had 98–99.3% sequence identity compared with the published gene sequences. This level of sequence identity may come about by sequence divergence between individuals within a species or they could be due to systemic ONT errors. For example, RbcL genes within the Diospyros genus have ~98.2% sequence identity [40]. However, much of the dissimilarity observed in the RbcL genes we extracted from our assemblies is likely to be due to systematic errors known to exist in ONT data: the most obvious being the inability of ONT to

accurately call the correct number of bases in homopolymer runs. Delahaye and Nicholis [41] analysed systematic issues associated with ONT long reads derived from bacterial and human samples and found that deletions occur at a frequency of 1.6–2.7% and are more likely to occur in GC rich regions. This may help to explain why we see up to 2% divergence in some of the RbcL genes, and indeed in the genomes themselves, here (i.e., the level of diversity is within the margin of error remaining in ONT-only assemblies). We are confident that *CLAW* can be used to help answer questions regarding plastid sequence evolution.

## Supporting information

**S1 File. Supplementary Alignments.** RbcL alignments from the algal, monocot, and dicot assemblies generated by *CLAW* and extracted from the reference genomes for each taxonomic group.
(DOCX)

**S1 Table. Information on ONT long reads used as input for *CLAW* and the Unicycler-generated chloroplast genome assembly statistics.**
(DOCX)

**S2 Table. Information on PacBio long reads used as input for *CLAW* and the Flye-generated chloroplast genome assembly statistics.**
(DOCX)

**S3 Table. Information on PacBio long reads used as input for *CLAW* and the Unicycler-generated chloroplast genome assembly statistics.**
(DOCX)

**S1 Fig. Neighbour-Joining tree of RbcL gene sequences extracted from reference chloroplast genome assemblies and those assembled by *CLAW*.** This tree is not meant to infer any phylogenetic relationships. Instead, we include it to show that reference and *CLAW*-assembled RbcL sequences are similar. Green, blue, and orange highlights indicate algal, monocot, and dicot species, respectively.
(TIF)

## Acknowledgments

We thank Chelsea Matthews for discussions about Snakemake workflows, which helped us improve on the quality of this work. We thank Professor Brian Atwell, and Dr. James Cowley for helping to review and improve this manuscript. AP acknowledges support from an Australian Research Training Program Scholarship and the FJ Sandoz Scholarship from the University of Adelaide.

## Author Contributions

**Conceptualization:** Aaron L. Phillips, Scott Ferguson.

**Data curation:** Aaron L. Phillips.

**Formal analysis:** Aaron L. Phillips, Scott Ferguson.

**Funding acquisition:** Rachel A. Burton.

**Investigation:** Aaron L. Phillips, Scott Ferguson.

**Methodology:** Aaron L. Phillips, Scott Ferguson.

**Project administration:** Aaron L. Phillips, Rachel A. Burton, Nathan S. Watson-Haigh.

**Resources:** Rachel A. Burton, Nathan S. Watson-Haigh.

**Software:** Rachel A. Burton, Nathan S. Watson-Haigh.

**Supervision:** Rachel A. Burton, Nathan S. Watson-Haigh.

**Validation:** Aaron L. Phillips, Scott Ferguson.

**Visualization:** Aaron L. Phillips.

**Writing – original draft:** Aaron L. Phillips, Scott Ferguson, Rachel A. Burton, Nathan S. Watson-Haigh.

**Writing – review & editing:** Aaron L. Phillips, Scott Ferguson, Rachel A. Burton, Nathan S. Watson-Haigh.

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
