## [Decision Letter · Decision Letter 0]

23 Oct 2023

Dear Phillips,

Thank you very much for submitting your manuscript "CLAW: An automated Snakemake workflow for the assembly of chloroplast genomes from long-read data" for consideration at PLOS Computational Biology.

As with all papers reviewed by the journal, your manuscript was reviewed by members of the editorial board and by several independent reviewers. In light of the reviews (below this email), we would like to invite the resubmission of a significantly-revised version that takes into account the reviewers' comments.

We cannot make any decision about publication until we have seen the revised manuscript and your response to the reviewers' comments. Your revised manuscript is also likely to be sent to reviewers for further evaluation.

Sincerely,

Christos A. Ouzounis

Academic Editor

PLOS Computational Biology

James O'Dwyer

Section Editor

PLOS Computational Biology

Reviewer's Responses to Questions

**Comments to the Authors:**

Reviewer #1: The authors developed a tool (CLAW) for automated assembly of chloroplast genomes from long read genomic sequences.

The tool is briefly described and the result of applying it to 19 public data sets is presented.

CLAW appears to be a useful tool, extending the set of available chloroplast assemblers with an option to use long read sequences.

I was able to install the software locally and run it on the provided example dataset.

Furthermore, I could apply it to a different dataset (reads: ERR3638927, reference: NC_035417.1) without problems (no full chloroplast was assembled using my settings, though).

I have some concerns that remain to be addressed by the authors.

Major concerns:

- the claim in the abstract (and again in l.136f) "currently no long-read, third generation tools exist" seems to be wrong. I found the recently published article "Plastid Genome Assembly Using Long-read data" presenting the tool ptgaul https://onlinelibrary.wiley.com/doi/10.1111/1755-0998.13787 (preprint published in late 2022)

- l.207: the recommendation to try different random seeds in case of failure to assemble the full chloroplast is a bit surprising to me. Does the random seed only affect the selection of reads for assembly? How seed dependent is success (how often did you observe success on a subsequent try)? How often should the user retry? Might there be a more sophisticated way to select a subset of reads (rather than randomly), that increases the chance for success? One such idea could be uniform coverage across the reference chloroplast.

- l.229: percent identity does not take large structural differences into account. Have the results been checked for structural variations, as well? The large size differences mentioned in l.292f seem to indicate, that this could be a problem.

- How exactly have the 19 test species/datasets been selected? Have any datasets been tested but excluded from the analysis? Have any of them be re-run with different seeds? If so, how often?

- Why have no PacBio datasets been used? This leaves the claim, that CLAW also works on PacBio data (l.349f) unsupported. Also, the speculation in l.372f could possibly be substantiated, if PacBio datasets were tested, as well.

- Given the relatively low sequence identity of 98.2% (l.407), compared to 99.9% with short reads (l.366), the possibility of using hybrid approaches with short- and long-read data or using short reads to correct long-reads should be discussed.

- l.415: why are you confident in using CLAW to answer questions regarding plastid evolution, given the high divergence due to systematic sequencing errors? This probably distorts phylogenetic analyses, particularly of closely related species or when genomes were sequenced with different technologies.

Minor concerns:

- l.74-81: the author summary contains a number of unproven, subjective or bold claims: existing tools being hard to install (which existing tools are you referring to here), the tool being easy to install and easy to use, the anticipation that barriers for entry to the field of bioinformatics are removed. These claims should be phrased a bit more cautious, or evidence should be provided.

- l.130: not necessarily the most frequent, but k-mers within a specific frequency range.

- l.153f: the default is automatic download from NCBI, but the reference can also be provided manually (documented in the README). Therefore, there is no need to discuss potential problems because of HPC configuration. Also, other concerns, like using an unpublished sequence as reference, do not arise for the reader.

- l.167: which mapper is used?

- l.168: shorter than a user-defined threshold (default: 5 kbp)

- l.169: why are reads larger than the expected chloroplast genome size excluded? They could contain multiple copies of the circular genome.

- l.186: in case of a complete circular chloroplast, is the sequence oriented and linearized consistently, e.g. always in order LSC, IR, SSC, IR?

- l.216: why have they been previously downloaded? Downloading them is part of the pipeline.

- l.251: not shown? Consider adding the alignments as a supplement.

- l.299f: how much difference has been observed in the IR regions? A circular assembly graph is also possible in case of identical IRs, e.g. if the arrangement of LSC and SSC could be resolved with spanning reads. (Same comment for l.319)

- l.352f: comparison of computation times on different hardware are problematic, this fact should be acknowledged.

- l.363: how was completeness defined and tested?

- consider deposition of the source code at Zenodo in addition to GitHub (so the software is archived and receives a dedicated doi).

- in the installation instructions in the README "cd long-read-chloroplast-assembly" should be replaced by "cd CLAW".

- consider replacing "--profile profiles/slurm" with "--cores 4" in the instructions, this should work on most systems, while the slurm profile requires a specific setup.

Reviewer #2: The authors present a pipeline to assemble chloroplast genomes using long reads. Their approach involves capturing reads from whole-genome sequencing projects and assembling them using two different tools. Additionally by making use of the snakemake workflow the steps are reproducible and easy to automate.

The manuscript is well written and easy to follow, however I have some concerns. The main one is that there is no comparison regarding performance with ptGAUL (Zhou et al, 2023) given that their methodology is similar. I would like to see a comparison of the performance of both tools and advantages and disadvantages of using each (advantage of snakemake workflow, flye vs flye & unicycler, circularization of the reference, polishing, etc).

Additionally I have some other questions:

1) Why use Unicycler? It seems that in your tests Flye gave better results. Do you have a hypothesis as to why more mitochondrial contigs are obtained using Unicycler?

2) For Figure 2 I would suggest adding the blue, red and green arrows to all 4 subfigures and not only in the first one to make them more comparable.

3) Was a test ran using the same individual for reference and reads? In this case obtaining a 100% identity was possible?

4) Regarding the github describing the tool, while I find it well explained I believe it can be improved to make it more user friendly for non specialized users. Additionally including a snakemake profile for workstations (or other non cluster configurations) might make the tool more accesible and easy to use for users without much experience with snakemake.

Reviewer #3: Phillips et al. have developed an automated workflow to help assemble chloroplasts from whole genome shotgun long read sequencing data. The workflow includes baiting reads and assembly either with Unicycler or Flye. The authors also describe potential issues that users should be aware of, such as the higher sequencing error rate using nanopore reads, and how inverted repeats, etc. can affect the assemblies and be seen in visualization of the assembly graphs via Bandage.

In general, I found the manuscript to be well-written and a good effort to fill in a need to assemble chloroplast genomes with long read sequencing data. There are various tools (as the authors cite) for short read data assemblies, but chloroplasts provide a particular challenge with genomic inversions and long read data can help with these issues. I think this study is a valuable contribution to the field.

Major comments:

1. The manuscript would benefit from a figure outlining the workflow in a graphical format.

2. The workflow only supports use on a cluster. There are still many labs that do not use clusters but run bioinformatics tools on their own servers and do not need a workflow that accommodates SLURM scheduling. Please include an alternative workflow such as this – it will help improve the reach of this tool and also encourage novice users as the authors state in the Author Summary section.

3. The manuscript would benefit from including PacBio test data for a few samples, or attempting to use a tool like Medaka to error correct. I appreciate that the authors point out the issues with the error rate in nanopore data, so it is unclear why they did not attempt to use PacBio data as well, or use medaka or to attempt to error correct their assembled genomes. Please explain or provide further analysis.

Minor comments:

1. Amend references to plants to at least algae, plants and other photosynthetic organisms. In certain places in the manuscript, the authors describe chloroplasts as being from plant cells or describe CLAW in respect to plants. Technically algae are not considered plants, such as diatoms and euglena, that have chloroplasts. Cyanobacteria also have chloroplasts. CLAW would be useful for assembling chloroplasts from these organisms as well. Please amend language in the manuscript to reflect this (e.g. line 42, 80)

2. Minimap2 is not stated as the method for read baiting and default parameters are not described in the methods overview. Please include in the methods section.

3. Line 149: authors mention fast5, POD5 is now the default and should also be mentioned

4. Line 292: Were all of the reference genomes circular? Illumina only genomes tend to be smaller than complete, circular genomes.

5. Error on the github installation directions, need to cd into CLAW, not “long-read-chloroplast-assembly”

**Have the authors made all data and (if applicable) computational code underlying the findings in their manuscript fully available?**

Reviewer #1: Yes

Reviewer #2: Yes

Reviewer #3: Yes

PLOS authors have the option to publish the peer review history of their article (what does this mean?). If published, this will include your full peer review and any attached files.

Reviewer #1: **Yes: **Markus J. Ankenbrand

Reviewer #2: No

Reviewer #3: No
---

## [Decision Letter · Decision Letter 1]

16 Jan 2024

Dear Dr Phillips,

Thank you very much for submitting your manuscript "CLAW: An automated Snakemake workflow for the assembly of chloroplast genomes from long-read data" for consideration at PLOS Computational Biology. As with all papers reviewed by the journal, your manuscript was reviewed by members of the editorial board and by several independent reviewers. The reviewers appreciated the attention to an important topic. Based on the reviews, we are likely to accept this manuscript for publication, providing that you modify the manuscript according to the review recommendations.

Sincerely,

Christos A. Ouzounis

Academic Editor

PLOS Computational Biology

James O'Dwyer

Section Editor

PLOS Computational Biology

Reviewer's Responses to Questions

**Comments to the Authors:**

Reviewer #1: The authors updated the manuscript, addressing most of the concerns raised by the reviewers. Importantly, the related published tool ptGAUL is now mentioned. Overall, the manuscript is hugely improved. My remaining concerns are:

- Evaluation of the completeness and accuracy of chloroplast genome assemblies is challenging, so thanks to the authors for providing some further detail on differences between CLAW assemblies and the reference. As mentioned in l.317ff, there are on average differences of more than 15kbp between assembly and reference. Even if some of the difference is due to an additional or missing copy of the IR, I would consider this inaccurate or incomplete. Thus, I don't think the claim in line 405, that CLAW was able to "assemble complete and accurate" genomes for all test datasets, is fully justified.

- Thanks for providing some RbcL alignments. However, all pairwise alignments should be shown, rather than a manual selection. Providing them all is easier to explain and does not raise questions about the selection procedure. Also, if they are provided in a suitable format (e.g. a zip of fasta files), the file size will be neglectible.

Reviewer #2: The new version of the submitted manuscript is greatly improved and I also believe that the addition of the diagram regarding the pipeline was an important addition. The authors have addressed all the issues and answered all the questions that I had.

Reviewer #3: I am mostly satisfied with the authors edits. I have two requests for minor revisions:

1. Change the last line (477) “Users may consider the use of short read technology, if available, to correct errors in their assemblies” to “Users may consider the use of short read technology, if available, or nanopore reads, if there was enough sequencing depth, to correct errors in their assemblies.” Or something similar. The reference cited also mentions Medaka, which uses nanopore reads to polish the nanopore assemblies, not just short read polishers. This also points out to the read that they may not need to generate another dataset for polishing.

2. Move the edited sentence from #1 up to the bottom of the paragraph describing the error rates that ends on line 434. I find that users new to ONT and PacBio are often unaware of the ability to polish assemblies, so moving that sentence and reference closer to the discussion I think will help reduce confusion for those readers.

Other than those two revisions, I recommend this manuscript for publication. An aside is that with the new Dorado basecaller from nanopore, polishing the error may soon be unnecessary with the 10.4.1 chemistry.

The authors were right to correct my comment about cyanobacteria and chloroplasts. That was a misunderstanding of the biology on my part.

**Have the authors made all data and (if applicable) computational code underlying the findings in their manuscript fully available?**

Reviewer #1: Yes

Reviewer #2: Yes

Reviewer #3: Yes

PLOS authors have the option to publish the peer review history of their article (what does this mean?). If published, this will include your full peer review and any attached files.

Reviewer #1: **Yes: **Markus J. Ankenbrand

Reviewer #2: No

Reviewer #3: No

Figure Files:

Data Requirements:

Reproducibility:

References:

---

## [Decision Letter · Decision Letter 2]

29 Jan 2024

Dear Dr Phillips,

We are pleased to inform you that your manuscript 'CLAW: An automated Snakemake workflow for the assembly of chloroplast genomes from long-read data' has been provisionally accepted for publication in PLOS Computational Biology.

Best regards,

Christos A. Ouzounis

Academic Editor

PLOS Computational Biology

James O'Dwyer

Section Editor

PLOS Computational Biology

Reviewer's Responses to Questions

**Comments to the Authors:**

Reviewer #1: The authors made all requested changes. The supplement now contains all rbcL alignments. While docx is not the ideal format, it is acceptable. However, one of the alignments (C. variabilis) is pasted as an image, rather than text. This should be corrected.

Reviewer #3: Thanks to the authors for addressing my final comments. I recommend this manuscript for publication.

**Have the authors made all data and (if applicable) computational code underlying the findings in their manuscript fully available?**

Reviewer #1: Yes

Reviewer #3: Yes

PLOS authors have the option to publish the peer review history of their article (what does this mean?). If published, this will include your full peer review and any attached files.

Reviewer #1: **Yes: **Markus J. Ankenbrand

Reviewer #3: No

---

## [Editor Report · Acceptance letter]

6 Feb 2024

PCOMPBIOL-D-23-00916R2 

CLAW: An automated Snakemake workflow for the assembly of chloroplast genomes from long-read data

Dear Dr Phillips,

I am pleased to inform you that your manuscript has been formally accepted for publication in PLOS Computational Biology. Your manuscript is now with our production department and you will be notified of the publication date in due course.

With kind regards,

Anita Estes
